# LKB1 and Tumor Metabolism: The Interplay of Immune and Angiogenic Microenvironment in Lung Cancer

**DOI:** 10.3390/ijms20081874

**Published:** 2019-04-16

**Authors:** Laura Bonanno, Elisabetta Zulato, Alberto Pavan, Ilaria Attili, Giulia Pasello, PierFranco Conte, Stefano Indraccolo

**Affiliations:** 1Medical Oncology 2, Istituto Oncologico Veneto IOV- IRCCS, 35128 Padova, Italy; alberto.pavan@iov.veneto.it (A.P.); ilaria.attili@iov.veneto.it (I.A.); giulia.pasello@iov.veneto.it (G.P.); pierfranco.conte@unipd.it (P.C.); 2Immunology and Molecular Oncology Unit, Istituto Oncologico Veneto IOV- IRCCS, 35128 Padova, Italy; elisabetta.zulato@iov.veneto.it (E.Z.); stefano.indraccolo@iov.veneto.it (S.I.); 3Department of Surgery, Oncology and Gastroenterology, Università degli Studi di Padova, 35128 Padova, Italy

**Keywords:** LKB1, tumor metabolism, tumor microenvironment, treatment response, lung cancer, immunotherapy, tumor angiogenesis

## Abstract

Liver kinase B1 (*LKB1*) is a tumor suppressor gene whose inactivation is frequent in different tumor types, especially in lung adenocarcinoma (about 30% of cases). LKB1 has an essential role in the control of cellular redox homeostasis by regulating ROS production and detoxification. Loss of LKB1 makes the tumor cell more sensitive to oxidative stress and consequently to stress-inducing treatments, such as chemotherapy and radiotherapy. LKB1 loss triggers complex changes in tumor microenvironment, supporting a role in the regulation of angiogenesis and suggesting a potential role in the response to anti-angiogenic treatment. On the other hand, LKB1 deficiency can promote an immunosuppressive microenvironment and may be involved in primary resistance to anti-PD-1/anti-PD-L1, as it has been reported in lung cancer. The aim of this review is to discuss interactions of LKB1 with the tumor microenvironment and the potential applications of this knowledge in predicting response to treatment in lung cancer.

## 1. Introduction

Cell processes, such as metabolic balance, DNA integrity maintenance, proliferation, polarity establishment, and interaction with tissue milieu, are essential for cell survival. For this reason, all these processes are finely tuned by multiple molecular pathways. Consistently, impairment of these processes and dysregulation of the control mechanisms are involved in neoplastic transformation.

Liver kinase B1 (*LKB1*, also known as *STK11*) is a tumor suppressor gene encoding a serine/threonine kinase of the calcium calmodulin family, ubiquitously expressed in several tissues and highly conserved among eukaryotes and involved in all these essential cell processes [1].

LKB1 has essential functions in the regulation of cell metabolism and acts at different levels. Tumor suppressor function of LKB1 is mainly mediated by the downstream AMP-activated protein kinase (AMPK). AMPK is a central metabolic sensor that controls glucose and lipid metabolism, coordinating a metabolic switch from anabolism towards catabolism under energy deprivation, such as glucose deprivation or hypoxia [1,2]. In addition, in cases of DNA damage, it is able to localize into the nucleoplasm and preserves cells from genomic instability; in this context, it directly interacts with ataxia telangiectasia mutated kinase (ATM) and actively cooperates with BRCA1 and the homologous repair (HR) machinery [3]. Moreover, LKB1 can also stimulate p53 activity and interfere with the expression of cyclins and cyclin-dependent kinase inhibitors, arresting cell cycle at G1 phase [4,5].

As far as cell polarity is concerned, preclinical models have shown how LKB1 may be crucial in its regulation: depletion of LKB1 disrupts cytoskeletal remodeling, as uncontrolled downstream microtubule affinity-regulating kinases (MARKs) lead to dynamic instability of cellular microtubules [6].

More recently, a correlation between LKB1 impairment and the presence of immune-suppressive microenvironment was observed [7,8].

LKB1 was originally identified as the causal mutation in Peutz–Jeghers Syndrome, a rare inherited disorder, characterized by benign gastrointestinal hamartomas, and by risk of developing cancer, that may be up to ten folds that of the general population [9]. Somatic mutations or loss-of-function alterations of *LKB1* are found in several malignancies, such as non-small cell lung cancer (NSCLC), cervical carcinoma, breast cancer, pancreatic cancer and melanoma [10,11,12,13,14,15,16]. In particular, in NSCLC *LKB1* gene is altered up to 30% of all cases, representing one of the most commonly mutated genes in these tumors and epigenetic events of gene inactivation have also been described. In this context, loss-of-function alterations may be present in up to 50% of patients with NSCLC, mainly involving non-squamous histology.

Even though *LKB1* genetic and epigenetic alterations in cancer have been known for long, their impact on prognosis is still controversial. Recently the role of its impairment has been investigated among patients carrying *KRAS* mutations: loss of LKB1 defines a distinct phenotype of NSCLC, characterized by an aggressive behavior and resistance to standard treatment [17].

The aim of the present review is to discuss how LKB1 and its multiple downstream pathways are able to influence neoplastic development and response to different kinds of treatments, given its master role in controlling cell interactions with tissue microenvironment.

## 2. LKB1 and Tumor Metabolism

The identification of LKB1 as the critical upstream kinase required for the activation of AMPK provided a direct link between tumorigenesis and cell metabolism [18,19,20].

AMPK is a central metabolic sensor found in all eukaryotes, which coordinates several key factors involved in multiple pathways to control energy homeostasis. In response to environmental alterations in nutrients and intracellular energy level, AMPK coordinates activation of catabolic pathways with the concomitant inhibition of anabolic processes, to maintain the steady-state levels of intracellular ATP [1].

Given its role as master regulator of cell metabolism, understanding of LKB1-AMPK activity and of its downstream pathway provides the possibility to identify new target and therapeutic strategies for cancer therapy.

### 2.1. Inhibition of Anabolic Pathways

The inhibition of anabolism under conditions of energy shortage is intended to minimize ATP consumption. AMPK was originally defined as the inhibitory upstream kinase for the critical metabolic enzymes Acetyl-CoA carboxylase (ACC) [21] and HMG-CoA reductase [22], which catalyze the first step in de novo lipid synthesis and the rate-limiting step for sterol synthesis, respectively, in a wide-range of eukaryotes. Fatty acid synthesis has been demonstrated to be essential for cancer cell survival, and its chemical inhibition has been associated with tumor growth suppression in prostate and lung xenografts [23,24]. This allows to speculate that inhibition of lipogenesis by LKB1-AMPK is an important aspect mediating its tumor suppressive role [24].

Under selective pressure, AMPK prevents the storage of glycogen by inhibitory phosphorylation of the glycogen synthases [25], and, importantly, its activation reprograms metabolism through transcriptional regulation of biosynthetic pathways. Moreover, through the phosphorylation of the key transcriptional regulators SREBP1 (sterol regulatory element-binding protein 1) [26], ChREBP (carbohydrate-responsive element-binding protein) [27] and HNF4α (hepatocyte nuclear factor 4α) [28], AMPK also controls lipid and glucose metabolism at the transcriptional level.

Through the inhibition of fatty acid synthesis and the activation of fatty acid oxidation, LKB1-AMPK axis plays a pivotal role in the maintenance of intracellular NADPH levels, which is required to prevent oxidative stress and to promote cancer cell survival under energy stress conditions [29].

Under cellular energy stress, LKB1-AMPK axis acts as a metabolic checkpoint inhibitor of cellular growth and inhibits protein synthesis, at least in part through the modulation of the mammalian target of rapamycin complex 1 (mTORC1) pathway [1,18]. AMPK also directly inhibits protein synthesis by phosphorylation of eEF2K [30], a negative regulator of protein elongation, which is a downstream target of the mTOR pathway [31]. Hypoxia-inducible factor 1a (HIF-1α) is a transcription factor regulated by mTORC-1. Hyper-activation of mTORC-1 promotes HIF-1α protein levels [32] and its downstream targets, such as vascular endothelial growth factor A (VEGF-A) and angiopoetin 2 (ANG2), several glycolytic enzymes and members of GLUT transporters. Thus, HIF1α activation in tumor contributes to Warburg effect [33,34], the propensity of tumor cells to utilize glycolysis instead of oxidative respiration even under normoxic conditions. Consequently, the LKB-AMPK axis acts as a negative regulator of the Warburg effect and suppresses tumor growth [32].

### 2.2. Activation of Catabolic Pathways

In order to replenish ATP levels, AMPK activation results in the stimulation of catabolic pathways. Among these, AMPK activation causes TXNIP13 and TBC1D1 phosphorylation and consequently increases the plasma membrane localization of the glucose transporters GLUT1 and GLUT4. This results in increased glucose uptake into tumor cells. In addition, AMPK indirectly increases glucose uptake through the phosphorylation of phospholipase D1 and actively modulates glycolysis through the phosphorylation of phosphofructo-2-kinase (PFK2), which affects the activity of PFK1, a rate-limiting enzyme in glycolysis [24,35].

In addition to glucose utilization, AMPK promotes mobilization of lipid stores, by stimulating lipases, such as adipose triglyceride lipase, to release fatty acids from triglyceride stores [36]. Free fatty acids are imported into mitochondria, where β-oxidation is stimulated following phosphorylation and inhibition of ACC activity by AMPK [21].

LKB1-AMPK-stimulated pathways also include increased turnover of macromolecules by autophagy [36]. Autophagy is a process by which cellular component, such as proteins, macromolecules, organelles and pathogens are recycled by specialized mechanisms, and it allows the turnover of old and damaged molecules, or the replenishment of nutrient stores under starvation. The ability of AMPK to directly phosphorylate Unc-51-Like Kinase 1 (ULK1) [37], a kinase essential for autophagy induction, establishes a direct link between LKB1-AMPK action and the autophagy process. Moreover, upon energy stress conditions, AMPK-mediated mTORC1 inactivation releases the inhibitory phosphorylation on ULK1 and other components involved in the autophagy [38]. Interestingly, AMPK may also directly phosphorylate members of the FOXO family of transcription factors, which regulate genes involved in autophagy [36].

## 3. LKB1 and Angiogenesis

Several studies indicate that LKB1 could be involved in the regulation of both physiological [30] and pathological [39] angiogenesis. As above mentioned, through its inhibitory function on mTOR signaling, LKB1-AMPK regulates HIF1α protein levels [33], and its downstream targets, such as VEGFA and ANG2 [34]. Consistently, in tumor angiogenesis models, cancer cells overexpressing LKB1 have reduced angiogenic activity compared with cells lacking LKB1 [39] (Figure 1). LKB1 is able to suppress the expression of VEGF, bFGF, MMP-2, and MMP-9 [39]. Moreover, mice carrying a targeted disruption of LKB1 die at mid-gestation with embryos and showed neural tube defects, mesenchymal cell death, and vascular abnormalities [30]. These phenotypes were associated with tissue-specific deregulation of VEGF expression, including a marked increase in the amount of *VEGF* mRNA [30].

LKB1 also functions as a RAB7 effector and suppresses angiogenesis by promoting the cellular trafficking of neuropilin-1 from the RAB7 vesicles to the lysosomes for degradation in lung cancer cells [40]. Neuropilin-1 is a VEGF receptor that can be expressed both by endothelial and tumor cells [41]. Its expression in tumors is correlated with increased microvessel density and enhanced tumor angiogenesis.

In addition, we recently demonstrated that LKB1 acts as a suppressor of NADPH oxidase 1 (*NOX1*) expression at transcript levels [42]. NADPH oxidases catalyze the transfer of one electron from NADPH to oxygen, generating superoxide or H_2_O_2_, thus resulting in increased oxidative stress [43]. High levels of redox oxygen species (ROS) have been involved in induction of angiogenesis and tumor growth [44]. Moreover, NOX1 has been associated in the promotion of angiogenic switch by a mechanism involving ROS generation and increased expression of VEGF [45]. Interestingly, genetic and pharmacological inactivation of NOX1 activity prevents the angiogenic switch and the growth of experimental tumors derived from NSCLC LKB1-deficient cancer cells [42].

Altogether, preclinical studies support the hypothesis that LKB1 triggers complex changes in the tumor microenvironment involving different targets that impact directly or indirectly on VEGF signaling and possibly other additional angiogenic pathways [42] (Figure 1).

## 4. LKB1 and Response to Non-Immunotherapy Treatments

### 4.1. Platinum-Based Chemotherapy

Different studies demonstrated that LKB1 is involved in the DNA damage response (DDR) machinery [46]. Upon induced DNA damage, LKB1 is phosphorylated at a particular site (threonine 363), which enables LKB1 to interact with ATM and co-localize with the latter to DNA damage foci [47]. Moreover, cell lines with a knocked down *LKB1* showed a reduced efficiency in the HR machinery, sensitizing cells to DNA damaging treatments, such as platinum compounds [3]. Similar to *BRCA1/2* mutated status, in preclinical models, LKB1 deficiency leads also to cell susceptibility to PARP inhibitors, exploiting the phenomenon of synthetic lethality. PARP-1 binds to single strand DNA breaks (SSBs) and is involved in the base excision repair (BER) pathway of DDR [48,49]. A drug-induced PARP-1 inhibition or downregulation leads to an accumulation of SSBs, that become double strand DNA breaks (DSBs) at replication forks: cells deficient in HR components, such as in the presence of non-functional LKB1, are not able to correct DSBs and die consequently [3].

LKB1 loss in NSCLC cells was recently associated with glutathione deficiency under oxidative stress and with sensitivity of cancer cells to cisplatin and γ-irradiation [50], supporting the hypothesis that LKB1-deficient tumors could be targeted by therapies inducing oxidative stress.

Consistently, in a retrospective analysis, improved overall survival (OS) was observed among patients treated with platinum-based chemotherapy and expressing low-null levels of LKB1 protein, confirming that dysfunctional LKB1/AMPK axis may impair the capacity of tumor cell to survive following platinum-induced DNA damage [51]. In line with these results, previous preclinical findings also correlate the pathway with the response to radiation-induced DNA damage [3]. Notably, LKB1-deficient lung cancer cells can also carry mutations in other genes involved in the regulation of oxidative stress, such as NRF2/KEAP1 [52]. Whether co-mutated tumors disclose a particular vulnerability to platinum compounds or radiation should be investigated in future studies.

More recently, Seo and colleagues retrospectively studied tumor samples of advanced NSCLC patients treated with first line platinum-based chemotherapy, in order to assess the potential role of single nucleotide polymorphisms (SNPs) associated with cancer-related pathways [53]. Ninety-five SNPs were found to be significantly associated with clinical outcomes, but only rs10414193A>G was consistently linked to worse chemotherapy response. This SNP increases LKB1 expression by increasing promoter activity, favoring cancer cell survival under chemotherapy-induced stress conditions, and causing resistance to cytotoxic agents [53].

### 4.2. Anti-Angiogenic Treatment

Preclinical evidence also supports a role of LKB1/AMPK axis in the response to anti-angiogenic treatment, mainly mediated by its ability to counteract metabolic stress [54,55].

On these preclinical bases, LKB1 role as a metabolic master regulator was studied as potential predictor of response to anti-angiogenic treatment in a retrospective study performed by our group [51]. Tumor samples of advanced NSCLC patients either treated with chemotherapy alone, or with chemotherapy and bevacizumab were analyzed for LKB1 expression using immunohistochemistry (IHC). Patients with low-null LKB1 protein levels had no significant benefit from the addition of bevacizumab. On the contrary, patients expressing moderate/intense levels of LKB1 protein had significant lower risk of death when treated with bevacizumab and chemotherapy, rather than with chemotherapy alone. To further investigate the impact of LKB1 on the response to anti-VEGF therapy, *LKB1* mutated and wild-type patient-derived xenografts (PDX) were generated and treated with bevacizumab. Loss of LKB1 was associated with reduced AMPK activation in LKB1-deficient tumors following anti-VEGF treatment, suggesting impaired ability to control the metabolic stress caused by this anti-angiogenic drug [54]. Moreover, bevacizumab administration was associated with significant development of larger necrotic areas in *LKB1*-mutated PDX [51]. It is interesting to note that necrosis often leads to recruitment of macrophages and other immune cells, which can promote angiogenesis and are associated with tumor escape from the VEGF blockade [56].

### 4.3. Other Non-Chemotherapy Agents

Liu and colleagues reported that *LKB1*-null lung cancer cell lines carry a much higher DNA damage rate and showed an increased dependence on Chk1 function [57]. Chk1 is a critical cell-cycle checkpoint kinase, which may halt the cell cycle during DDR or lead to apoptosis if the damage is unrepairable [58]. *LKB1*-null cells showed higher levels of Chk1 and revealed tumor sensitivity to Chk1 inhibition both in vitro and in vivo. In particular with Chk1 inhibitor AZD7762 led to a reduced metabolic activity in mice carrying *LKB1* deficient lung tumors, but this did not translate into size reduction. Strikingly, the combination of AZD7762 with a DNA synthesis inhibitor like gemcitabine was able to cause a significant tumor regression [57].

*LKB1* mutations enhance also cell sensibility to the inhibition of WEE1 kinase, another cell-cycle checkpoint [59]. When activated, WEE1 causes direct inhibition of CDC2/cyclin B-mediated cell cycle progress from G2 to M phase [60,61]. Pharmacological inhibition of WEE1 activation has been shown to have antitumor effects, by enhancing genotoxic effects of standard chemotherapy and reducing chemoresistance-potential of tumor cells [62]. In an experimental model, human NSCLC cell lines carrying LKB1 inactivation were significantly more sensitive to the action of a WEE1 kinase inhibitor, AZD1775, compared with the ones expressing LKB1. Likewise, mice models of LKB1-null NSCLC benefitted from the addiction of AZD1775 to cisplatin in terms of median OS, compared to wild-type models. Interestingly, LKB1 proficient tumor cells, carrying an LKB1 variant not able to interact with ATM, were sensitive to AZD1775 as LKB1-null ones, confirming the essential role of LKB1-ATM direct cooperation for an effective DDR system [59].

These studies support a potential role of LKB1 status in the response to a wide range of new small molecules, not yet available in the clinical setting.

Another field that deserves further investigations concerns the role of LKB1 in *EGFR*-mutated NSCLC. Even though inactivating mutations of *LKB1* are rare in *EGFR*-mutated tumors [63], recent studies demonstrate a new mechanism of LKB1 inactivation, mediated by beta2-adrenergic receptor stimulation of protein kinase C, which in turn phosphorylates LKB1 at serine 428 and inhibits it [64,65]. This particular pathway linked to stress hormone signaling seems to be implicated as a driver of the development of T790M-independent EGFR tyrosine kinase inhibitors (TKI) resistance [66]. Investigation about its potential role in the development of resistance to third-generation EGFR-TKI and in the sensitivity to the association of anti-EGFR and bevacizumab is awaited.

The role of LKB1 in addressing tumor response to metabolic stress [54,67] also opens new perspectives for “old drugs”. Recently, Moro and colleagues performed a study on the usage of metformin with cisplatin in co-mutated *KRAS/LKB1* NSCLC [68]. Metformin causes the inhibition of complex 1 of the electron transport chain, leading to a decrease of intracellular ATP levels and inducing energetic crisis [69]. *KRAS/LKB1* co-mutated NSCLC cells showed a metabolic frailty and this could be exploited introducing drugs that are able to induce high metabolic stress, such as metformin. Low doses of metformin significantly enhanced cisplatin-induced apoptosis only in NSCLC cell lines with *LKB1* deletion. According to in vitro results, *KRAS/LKB1* co-mutated PDXs showed meaningful tumor shrinkage upon metformin combination treatment with cisplatin. Moreover, in *KRAS/LKB1* mutated PDX model, metformin was able also to prevent the development of acquired tumor resistance to chemotherapy. A clinical trial is ongoing to test the hypothesis (Table 1).

## 5. LKB1 and Immunotherapy

### 5.1. LKB1 and Immune Microenvironment

In the latest years, immunotherapy has been introduced in the treatment algorithms for NSCLCs, given its proven efficacy in prolonging survival in different clinical settings [70,71,72,73,74]. However, comprehensive data about biomarkers of response to immunotherapy are lacking. While LKB1 role in tumor metabolism has been widely investigated, much less is known about its interaction with the immune microenvironment, even though recent studies have opened new perspectives in this field.

Few preclinical studies have been conducted to investigate LKB1 involvement with the immune system, considering its expression both on immune cells and on tumor cells.

Recently, a function of LKB1 in regulating the development, proliferation and activation of T effector and T regulatory cells (T reg) has been demonstrated. The inactivation of LKB1 in T cells is responsible for impaired T reg function through the lack of Foxp3 stabilization [75], and it is also related to blocked thymocyte development and reduction in the number of peripheral T cells [76]. However, the reduced pool of T effector cells showing loss of LKB1 is characterized by increased T cell activation and inflammatory cytokine production [76,77].

LKB1 also regulates NFkB-mediated macrophage activation: upon lipopolysaccharide (LPS) stimulation, LKB1-deficient macrophages show higher production and expression of pro-inflammatory cytokines, together with enhanced NFkB activity, compared to *LKB1* wild-type macrophages [78]. Interestingly, LPS stimulation induces selective LKB1 depletion in dendritic cells, leading to an expansion of T reg cells [79].

Koyama and colleagues have deeply investigated the effects of LKB1 on the tumor immune microenvironment studying *KRAS/LKB1* mutant (KL) NSCLC mouse models, human samples, and cell lines [80]. They found significantly increased number of tumor-associated neutrophils (TAN) and decreased number of tumor-associated macrophages (TAM) in KL tumors compared to tumors showing alterations only in *KRAS* (K) (Figure 1). Moreover, TAN in KL tumors showed higher expression of T cell suppressive factors and IL-1α, which is tumor-promoting through the induction of IL-6 and subsequent STAT3 phosphorylation [80,81]. These findings are consistent with a higher expression of pro-inflammatory and neutrophil-recruiting chemokines and cytokines, such as CXCL7, CXCL5, and IL-1α, and lower expression of lymphocytes and dendritic cells recruiting chemokines, such as CCL5 and CXCL12, in KL tumors compared to K tumors [80]. In addition, KL tumors showed lower number of tumor infiltrating lymphocytes (TILs), which resulted unbalanced in favour of T reg cells and expressed higher levels of T cell exhaustion markers. Finally, PD-L1 expression was significantly lower in KL than in K tumors, thus reflecting potential inefficacy of anti-PD-1/PD-L1 treatments [17,80].

More recently, the particular connection between LKB1 expression and the stimulator of interferon genes (STING) pathway was also studied [8]. STING is a cytosolic protein activated by the detection of free double-strand DNA into the cytoplasm, due to pathogen infection or to neoplastic transformation [82]. Aberrant cytoplasmic double-strand DNA activates STING and promotes its cellular relocation: it binds TKB1 (TANK-binding kinase 1), a regulator of innate immune signaling, and causes the expression of interferon and other chemokines, which stimulate T cell recruitment. In the context of cancer, STING may play a crucial role as one of the first steps in the immune surveillance process. Kitajima and colleagues identified the downregulation of STING in *KRAS*-*LKB1* mutated cancer cells. In particular, *LKB1* inactivation dysregulates serine metabolism, leading to increased levels of S-adenylmethionine (SAM); SAM, in turn, is substrate for multiple epigenetic silencing enzymes, such as DNMT1 and EZH2, that are both directly involved in the methylation of STING promoter, causing its down-modulation [8] (Figure 1).

Altogether these evidences suggest that tumors lacking LKB1 are endowed with multiple means to escape immune system. Consequently, a therapy based only on immune checkpoint blockade could have an intrinsic crucial limitation, encouraging the development of synergistic therapeutic strategies able to target cancer cells with such molecular features.

### 5.2. LKB1 and Response to Immunotherapy

Preclinical evidences, sustaining a potential involvement of LKB1 loss in inducing an immune-suppressive tumor microenvironment, have been confirmed by the clinical evidence that showed that has co-mutated *KRAS-LKB1* advanced NSCLC obtained limited benefits from immunotherapy [7]. A large retrospective cohort of *KRAS*-mutant NSCLC patients receiving immune checkpoint inhibitors (ICIs) has been studied: *KRAS*-mutant lung adenocarcinomas harboring *LKB1* mutations obtained inferior response rate, shorter progression free survival (PFS) and OS to PD-1 and/or CTLA-4 blockade, compared to *KRAS*-mutant lung adenocarcinoma without co-mutations and to those with *TP53* co-mutations. These results were validated in an independent group of patients enrolled in the Checkmate 057 trial, where LKB1 status was investigated also by using IHC and considering epigenetic alterations of the genes. In addition, even though KL NSCLCs have lower PD-L1 expression when compared with the other *KRAS* mutated patients, the presence of mutations resulted independent on tumor mutational burden score and the association of *LKB1* mutations with worse outcome was also confirmed among PD-L1 positive subgroup of KL NSCLC treated with ICIs. Overall, these data suggest a potential independent association of *LKB1* co-mutations with worse outcome to ICIs among *KRAS*-mutated patients, although the predictive role cannot be formally established due to the lack of a numerically adequate control group treated without ICIs [7].

In parallel, an independent study, considering a large retrospective series treated with ICIs and analyzed by genotyping, demonstrated that *LKB1* mutations are associated with lack of durable clinical benefit (defined as partial response or stable disease lasting more than six months) [83].

Taken together, preclinical and clinical findings suggest a role of LKB1 in sustaining a cold tumor immune microenvironment, which is responsible of primary resistance to treatments targeting the immune system, in particular ICIs. However, given the fact that *LKB1* loss seems to increase response to platinum-compounds, it remains to be investigated how mutations of this gene modulate response to combined ICIs-chemotherapy treatment, which is increasingly offered to NSCLC patients.

## 6. The Interplay between Immune and Angiogenic Microenvironment

As described above, several clinical and pre-clinical evidences support a role of LKB1 as modulator of tumor microenvironment, through direct influence on angiogenesis and on the immune landscape. Specifically, LKB1 loss directly or indirectly impacts on VEGF signaling and possibly on other additional angiogenesis pathways, leading to promotion of angiogenesis [42] and, in parallel, a lack of LKB1 activity within tumor cells is related to the creation of an immunosuppressive microenvironment [80] (Figure 1).

Angiogenic and immune microenvironments are not independent. VEGF, the main mediator of angiogenesis, is well known to have inhibitory effects on T cells [84,85]. Indeed, VEGF can modulate different classes of immune cells. It inhibits maturation of dendritic cells, impairing their role as antigen presenting cells, and it favors recruitment of myeloid-derived suppressor cells (MDSCs). As far as T lymphocytes are concerned, VEGF has a role both in the recruitment and in the proliferation of T reg cells at tumor site, upregulating the expression of immune checkpoints on cytotoxic T cells [84,86,87]. On the other hand, activation of the STING intracellular phosphorylation cascade leads to the increase of several immune inflammatory cytokines, including IFNβ, CXCL10, CCL5, GM-CSF, CCL3, and IL-1α and to the suppression of IL-6, as well as the activation of STAT1, which stimulates PD-L1 expression and antitumor innate immunity signals. IFNs and their downstream modulator STAT1 have been demonstrated to negatively regulate angiogenesis [88,89]. It was also reported that GM-CSF can inhibit breast cancer growth in mice models, reducing tumor angiogenesis [90], and CXCL10 is capable of the inhibition of the angiogenic process [91] (Figure 1).

Moreover, one of the known escape mechanisms through which tumors evade immune response concerns an impairment of the lymphocytes-endothelium interaction (Figure 2). Pro-angiogenic factors cause a reduction in the expression of the adhesion molecules and subsequent defect in the adhesion of immune cells to neo-formed vessels. Moreover, VEGF may also have a direct role of reducing the adhesion of lymphocytes, also independently on adhesion molecules [92].

The interaction between angiogenic and immune tumor microenvironment may be particularly relevant in specific clinical settings. Liver metastasis and hepatocellular carcinomas have increased levels of VEGF and the specific role of tumor angiogenesis has been studied and related to the immune-suppressive status [93,94].

The complex interplay between the angiogenic mediator VEGF and the immune microenvironment within the tumor reinforces the hypothesis of a potential central role of LKB1 status in the response to immunotherapy combination treatment [95].

## 7. Conclusions and Perspectives

The frequency of genetic and epigenetic alterations of LKB1 in NSCLC, in parallel with its role in cell metabolism and its interplay with several main cellular pathways, supports an important role of LKB1; therefore, loss of LKB1 leads to important deregulation in cellular homeostasis, gain of metastatic potential, and development of resistance to systemic treatment in lung cancer. In particular, while the central role of LKB1 in cell metabolism and modulation of tumor microenvironment is emerging, the first evidence is available about the potential role of its genetic and epigenetic alterations in determining responsiveness of NSCLC to cancer treatment [7,51]. These findings have great potential clinical applications and open new perspectives also in the study of tumor microenvironment.

The potential predictive role of LKB1 impairment in causing resistance to ICIs treatment is supported by interesting preclinical evidence and, if further confirmed, may potentially change the therapeutic approaches to KL NSCLCs.

Further insights on the role of this pathway in the response to ICIs and on the potential role of combination treatments in this setting may be provided also by further studies about the changes in immune-related and angiogenic tumor microenvironments induced by LKB1 impairment in clinical NSCLC samples. The co-existence of genetic and epigenetic alterations of *LKB1* in the presence of other genetic alterations associated to relative resistance to ICIs, such as *MET* mutations, has also not been studied extensively yet and may provide further information for a new era of personalized medicine integrating genetic information into clinical practice also for non-oncogene addicted NSCLC [96].

The involvement of LKB1 in modulation of tumor microenvironment, affecting both angiogenic and immune-related milieu may be particularly relevant in specific clinical settings.

Interestingly, the combination of anti-VEGF therapy and anti-CTLA-4 blockade has been associated with favorable clinical outcomes in patients with metastatic melanoma [97]. In parallel, first clinical evidence is available about a potential synergism between anti-angiogenic treatment and ICIs in lung cancer. The association of chemotherapy, anti-PD-L1 treatment and bevacizumab has demonstrated to be feasible and particularly effective in clinically selected subgroups of patients, such as patients with liver metastasis, who seem to have limited clinical benefit from the addition of immunotherapy to standard chemotherapy in first line setting [98]. Similarly, *EGFR*-mutated and *ALK*-rearranged patients seem to have increased benefit from the association of both anti-angiogenic treatment and ICI (in addition to chemotherapy) [98]. These data support the idea of an interplay between angiogenic and immune-related microenvironment and the need for further personalization of treatment in advanced NSCLC.

Overall, the role of LKB1 in lung cancer warrants further investigation and several open questions are still to be addressed.

From a technical point of view, we still do not know which is the best method to investigate “LKB1 impairment”. An association between genetic alterations and protein expression has been demonstrated by our group [51], but potential functional differences among different genetic and epigenetic alterations are not known yet. Potential reliability of liquid biopsy in detecting genetic alterations of *LKB1* by NGS is also an open question. As a matter of fact, genetic characterization of *LKB1* requires analysis of several exons and the amount of DNA required can be limiting in clinical practice. For this reason, the feasibility of routine assessment of the *LKB1* mutational status in clinical practice will require additional studies.

With respect to LKB1 as a predictive biomarker in advanced NSCLC patients, the main point to face is the confirmation of its predictive role in a study providing an adequate control group treated with chemotherapy (without ICIs).

It would be also interesting to understand if the role of LKB1 impairment is independent on *KRAS* mutations and may be extended to all NSCLC patients, even in the absence of *KRAS* mutations.

Data about the role of LKB1 in patients with combination immunotherapy are also not available. Evaluation of LKB1 status at baseline in tissue in future studies involving combination treatment, in particular with ICIs plus chemotherapy and ICIs plus anti-angiogenic agents, is suggested to provide further information for personalized combination treatment strategies.

Finally, no information is available about the role of LKB1 in lung metastases from other solid tumors, even though a role of LKB1 in the acquisition of metastatic potential of other cancers, such as triple negative breast cancer and HPV-cervical cancer has been studied [99,100].

In conclusion, evidence supports the importance of upfront analysis of LKB1 status in patients enrolled in clinical trials evaluating new therapeutic strategies in NSCLC and further knowledge about its role in the interplay between angiogenic and immune tumor microenvironments may help in studying new combination strategies in advanced NSCLC.

## Figures and Tables

**Figure 1 ijms-20-01874-f001:**
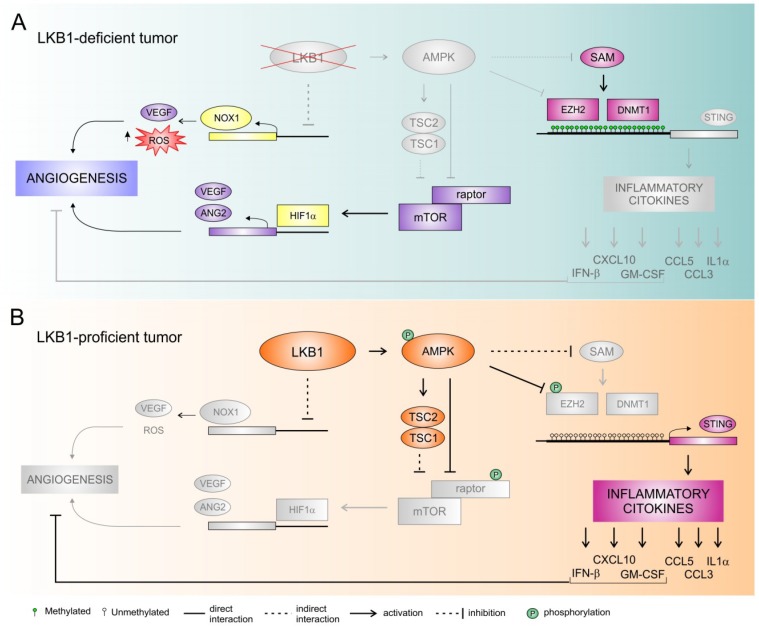
LKB1-mediated regulation of tumor angiogenesis and immune escape: a working model. (**A**). Loss of LKB1 is associated with increased expression of NADPH oxidase 1 (NOX1) transcript. NOX1 promotes the angiogenic switch by increasing redox oxygen species (ROS) generation and expression of vascular endothelial factor (VEGF). By triggering mTOR activity, lack of AMPK activation promotes increased expression of HIF-1α and of its downstream targets, such as VEGF and angiopoetin 2 (ANG2). Moreover, loss of LKB1 promotes serine utilization and synthesis of S-adenosyl methionine (SAM), a substrate for multiple epigenetic silencing enzymes such as DNMT1 and EZH2. This results in silencing of stimulator of interferon genes (STING) expression. STING inhibition determined reduction of PD-L1 expression and downregulation of chemokines that promote T-cell recruitment, facilitating immune escape. (**B**). LKB1 acts as suppressor of NOX1, and, through the activation of AMPK, inhibits mTORC1, by activating the negative mTORC1 regulator TSC2 and by inhibiting the mTORC1 subunit RAPTOR. This results in reduced expression of VEGF and angiogenesis. AMPK activation inhibits methylation of the *STING* promoter by methyltransferases DNMT1 and EZH2. Moreover, AMPK also directly phosphorylates and inhibits EZH2. Activation of STING intracellular phosphorylation cascade led to the release of the immune inflammatory cytokines such as IFNβ, CXCL10, CCL5, GM-CSF, CCL3, and IL1α, which leads to increased antitumor innate immunity signals and higher PD-L1 expression. Moreover, IFNβ, CXCL10, and GM-CSF may also contribute to inhibition of tumor angiogenesis.

**Figure 2 ijms-20-01874-f002:**
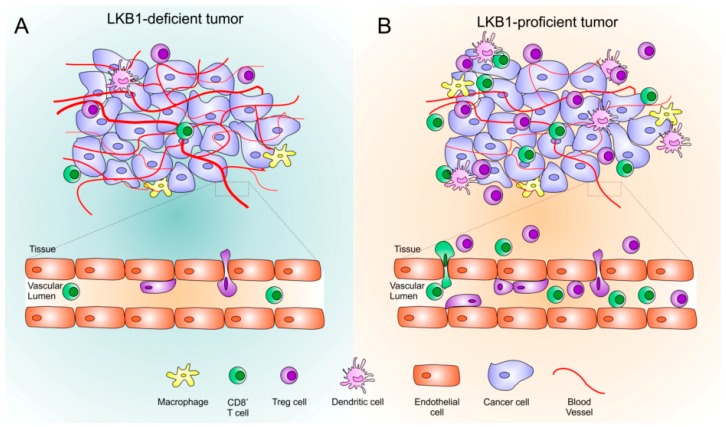
The LKB1-mediated interplay between immune and angiogenic microenvironment. (**A**). Loss of LKB1 in tumor cells directly or indirectly impacts on VEGF pathway, triggering tumor angiogenesis. Pro-angiogenic factors might cause the reduction of the adhesion molecules and the consequent defect in the adhesion of immune cells to neo-formed blood vessels. By coordinating different intracellular pathways, lack of LKB1 sustains a cold tumor immune microenvironment. (**B**). LKB1 regulates tumor angiogenesis, and is associated with a proficient immune-surveillance system: tumor infiltrating lymphocytes are well represented, especially the T effectors (T eff) subset, due to the release of T eff recruiting chemokines and diapedesis-permissive vasculature [92].

**Table ijms-20-01874-t001a:** 

A. Clinical Trials Requiring STK11/LKB1 Determination as Inclusion Criteria				
Clinical Trial ID	Phase	Brief Description	Study Population	Primary Endpoint
NCT03709147 (FAME trial)	II	Exploiting metformin plus/minus cyclic fasting mimicking diet (FMD) to improve the efficacy of platinum-pemetrexed chemotherapy in advanced LKB1-inactive lung adenocarcinoma	LKB1 inactive advanced NSCLC	PFS
NCT03872427 (BeGIN trial)	II	A phase II Basket trial of Glutaminase Inhibitor (BeGIN) CB-839 HCl in patients with NF1 aberrations, NF1 Mutant Malignant Peripheral Nerve Sheath Tumors (MPNST), KEAP1/NRF2 and LKB1 aberrant tumors.	NF1 aberrations, NF1 Mutant Malignant Peripheral Nerve Sheath Tumors (MPNST), KEAP1/NRF2 and LKB1 aberrant tumors	ORR
NCT03375307	II	A phase II study of Olaparib (AZD2281) in patients with metastatic/advanced urothelial carcinoma with DNA-repair defects.	Metastatic/advanced urothelial carcinoma with DNA-repair defects (among these: *STK11* gene mutations).	ORR
NCT02352844	II	A Phase II Study of Everolimus in Patients With Advanced Solid Malignancies With *TSC1*, *TSC2*, *NF1*, *NF2*, or *STK11* mutations.	Advanced solid malignancies with *TSC1*, *TSC2*, *NF1*, *NF2*, or *STK11* mutations.	ORR
NCT02645149	IV	Molecular profiling and matched targeted therapy for patients with metastatic melanoma. Once standard therapies have been exhausted, patients receive a targeted therapy matched for their genetic result, if applicable. If *STK11* mutated, they receive everolimus.	*BRAF* and *NRAS* wild-type unresectable Stage III or Stage IV metastatic melanoma.	Type and frequency of genetic aberrations in *BRAF*/*NRAS* wild-type metastatic melanoma and proportion of patients with BRAS/NRAS wild-type melanoma receiving targeted therapy

PFS: progression-free survival, ORR: overall response rate.

**Table ijms-20-01874-t001b:** 

B. Clinical Trials Investigating STK11/LKB1 Status Among Secondary Objectives				
Clinical Trial ID	Phase	Brief Description	Study Population	Primary Endpoint
NCT01470209	I	A phase I study assessing the safety of the combination of everolimus and BKM120 for the treatment of advanced solid tumors cancer in patients who are no longer benefiting from or unable to withstand standard treatment.	Solid tumors (including lung cancer). Alterations in *PIK3CA*, *NF1*, *TSC1*/*TSC2*, *mTOR*, *KRAS*, *LKB1*, *PTEN* will be accessed.	DLT
NCT02642042	II	A phase II Trial of Trametinib with docetaxel in Non-Small Cell Lung Cancer (NSCLC) *KRAS* mutated patients after one or two prior systemic therapies.	Advanced NSCLC carrying *KRAS* mutation Tertiary objective: To evaluate the response rate in the presence of co-mutations *TP53* and *LKB1*	ORR
NCT01310231	II	A randomized phase II, double blind trial of standard chemotherapy with metformin (versus placebo) in women with metastatic breast cancer receiving first, second, third or fourth line chemotherapy with anthracycline, taxane, platinum, capecitabine or vinorelbine based regimens.	Metastatic breast cancer in first, second, third or fourth line chemotherapy treatment. Immunohistochemistry analysis of different markers (IR, LKB1, phosphorylated AKT, S6K, ribosomal protein S6, 4E-BP1, and stathmin) performed.	PFS
NCT02285855	II	Tumor mutation status and metabolic response to metformin in non-small cell lung cancer (NSCLC).	NSCLC undergoing Stereotactic body Radiotherapy (SBRT). Genotype comparisons of J*KRAS*, *STK11*, and *TP53* mutations assessed.	ORR [Closed for poor accrual]
NCT03495544	Observational	Comparative multicenter study estimating association between germline DNA-repair genes mutations and PD-L1 expression level in breast cancer.	Breast cancer. Association between germline mutations (*TP53 MLH1 MSH2 MSH6 PMS2 EPCAM APC MUTYH CDKN2A CDK4 ATM KIT PDGFRA CDH1 CTNNA1 PRSS1 SPINK1 BRCA1 BRCA2 FANCI FANCL PALB2 RAD51B RAD51C RAD54L RAD51D CHEK1 CHEK2 CDK12 BRIP1 PPP2R2A BARD1 PARP1 STK11 XRCC3*) and PD-L1 expression.	Diagnostic performance of PD-L1 expression in breast cancer

DLT: dose limiting toxicity; PFS: progression-free survival, ORR: overall response rate.

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
