# Peer review of "LKB1 and Tumor Metabolism: The Interplay of Immune and Angiogenic Microenvironment in Lung Cancer"

_ijms, 2019, doi:10.3390/ijms20081874_

Reviewer 1 Report

This is an excellent review describing of the role of LKB1 loss and/or mutation in NSCLC patients response to ICIs and anti-angiogenics. It provides a comprehensive analysis of the recent preclinical and clinical data. Authors also documented on the complexity of mechanisms orchestrated directly or indirectly by LKB1 and its potential dependency on KARS mutation in lung cancer and response to ICIs.The review is of great interest for large audience and can have a great impact in the field. However, Figure 1 is missing from the PDF file and some missing aspects can be presented in an additional second figure. For instance how LKB1 is regulating HIF1alpha and VEGF in cancer cells and how  LKB1 regulates PD-1/PDL1 expression in cancer or inflammatory cells. Also a non-exhaustive list (in a format of table) of preclinical and clinical trials with ICIs, chemotherapy and the combination with anti-VEGF with the status of KRAS LKB1 of on NSCLC would be very helpful.

Author Response

We thank the reviewer for the comments.

We have  created a second figure (now called Figure 1)  on the role of LKB1 in influencing angiogenic microenvironment.

We have also created a table (Table 1) summarizing the clinical trials ongoing in solid tumors in which LKB1 status is evaluated as an inclusion criteria (A) or as a subgroup analysis (B).

Reviewer 2 Report

The authors present an elaborate review of LKB1 in the tumor microenvironment, in particular its role in lung cancer immunology and angiogenesis. They include the role of LKB1 in tumor metabolism, its role in tumor angiogenesis, its role in response to treatments dependent and independent of immunotherapies. The following comments are intended to improve the manuscript.

1)      Since the lungs are a frequent site for metastasis of other tumor types, could the authors provide any information known about whether the information reviewed on LKB1 in lung cancer is also applicable or studied in lung metastases from other tumor types.

2)      Could the authors present more detail about the major questions that need addressed regarding LKB1 in the near future.

Author Response

We thank the reviewer for the comment.

To the best of our knowledge there is no specific evidence available on the role of LKB1 in lung metastases from other tumor sites. Nevertheless, tumors with described impairment of LKB1 such as breast or cervical cancer may cause lung metastases and a specific role of LKB1 in the acquisition of metastatic potential has been studied. We commented about the issue in the text (last paragraph, page 9).

We have also included a new paragraph specifically addressing the issue of open questions that need to be answered in the next future (page 9).

Reviewer 3 Report

The authors did a great job with this review that describes the role of LKB1 signaling. I believe this review is well written and need no further suggestions.

Author Response

We thank the reviewer for the comment.